# Pilot Study of Electronic Nicotine Delivery Systems (ENDS) Cessation Methods

**DOI:** 10.3390/pharmacy9010021

**Published:** 2021-01-14

**Authors:** Michelle Sahr, Shelby Kelsh, Noah Blower, Minji Sohn

**Affiliations:** College of Pharmacy, Ferris State University, Big Rapids, MI 49307, USA; ShelbyKelsh@ferris.edu (S.K.); blowern@ferris.edu (N.B.); MinjiSohn@ferris.edu (M.S.)

**Keywords:** electronic nicotine delivery systems, vaping, cessation, nicotine addiction

## Abstract

Currently, 7.6% of the U.S. young adults aged 18–24 years old use e-cigarettes. This study piloted three methods of Electronic Nicotine Delivery Systems (ENDS) cessation by measuring cessation rates, motivational techniques that contributed to cessation success, and participants’ changes after decreasing vape use. Participants were randomized into three study arms (nicotine replacement therapy (NRT) + behavioral support, vape-taper + behavioral support, self-guided) in a 1:1:1 ratio. All participants were invited to attend nine in-person or phone appointments over the 6-month study period. At 12 weeks, 3 of 7 (42.9%) participants in the NRT + behavioral support arm, 6 of 8 (75%) vape-taper + behavioral support arm, and 7 of 9 (77.8%) self-guided arm self-reported being vape-free and nicotine-free. At 6 months, 3 of 7 (42.9%) participants in the NRT + behavioral support arm, 6 of 8 (75%) vape-taper + behavioral support arm, and 4 of 9 (44.4%) self-guided arm self-reported being vape-free and nicotine-free. A challenge to quitting and remain quit is social pressures, but participants identified self-control and establishing new habits to be the best methods to overcome the desire to vape. Participants who received behavioral support and a vape-taper plan from pharmacists were more likely to be vape-free and nicotine-free at 6 months.

## 1. Introduction

Electronic nicotine delivery systems (ENDS), commonly referred to as vapes or e-cigarettes, have gained popularity in recent years, especially among teens and young adults [1]. The most recent U.S. data reports 7.6% of 18–24-year-olds use electronic cigarettes and 1.7% of 18–24-year-olds are dual users of combustible cigarettes and electronic cigarettes [2]. Many adolescents try flavored ENDS products because of social pressures without knowing they contain nicotine, which can lead to nicotine addiction, increased use, and use of marijuana [3,4,5,6]. Adolescents are also more likely to transition to traditional cigarettes after trying ENDS. Furthermore, 30.7% of ENDS users reported recent use of at least one combustible tobacco product at a six-month follow-up, compared to 8.1% of participants who had never used ENDS [7].

While many people believe ENDS to be safer than combustible cigarettes, research has shown that ENDS users obtain plasma nicotine concentrations similar to combustible products and have been linked to many health issues [8,9,10,11,12,13]. Since 2019, e-cigarette or vaping product use associated lung injury (EVALI) has been diagnosed in over 2800 patients and the cause of 68 deaths [9].

Many healthcare providers are unsure of how to help their patients quit ENDS due to lack of research [10,11,12,13]. National Cancer Institute and the Truth Initiative each sponsor a tobacco quit service that utilizes text messaging for behavioral support [14,15]. The Truth Initiative’s BecomeAnEx program had 46% of participants report reduced ENDS use and 16% report ENDS cessation in two weeks. Most enrollees indicated a desire for additional and longer support in their quit attempt [16]. This research shows that behavioral support and frequent touchpoints are important components of quitting ENDS, but neither program assesses pharmacotherapy or sustained abstinence. The use of nicotine replacement therapy (NRT) has been well-established in smoking cessation for traditional cigarettes, but currently there is only one case report for using NRT in the cessation of ENDS, giving limited information on how to appropriately dose patients [17]. Additional literature to support clinicians to help ENDS users to quit is a case study using a vape-taper and another case study using varenicline [18,19].

The objectives of this study were to pilot three methodologies of ENDS cessation by measuring cessation success rates, motivational techniques that contributed to cessation success, and participants’ changes after decreasing vape use.

## 2. Materials and Methods

Adults who currently vape were recruited to enroll in the study through informational fliers posted in Big Rapids, MI, USA and emails sent through a university’s listserv to employees and students. The enrollment goal was 30 total participants with 10 in each arm. Researchers recruited from May 2019–January 2020 and enrolled participants on a rolling basis.

Inclusion criteria for the study population included adults who used ENDS at least four days a week and were motivated to quit within two weeks. Exclusion criteria included the following: participants who were pregnant or plan to become pregnant, have had a heart attack or stroke in the past two weeks, or have poorly controlled chronic obstructive pulmonary disease (COPD) or asthma. Participants were asked to refrain from using any nicotine products other than ENDS during the study period.

The framework of the study design was based off of smoking cessation research using a 12-week timeframe, high touch points with participants, and healthcare professionals offering support and guidance [20,21].

Block randomization was used to place eligible participants in one of three arms NRT (nicotine patch +/− nicotine lozenge or gum) + behavioral support, vape-taper + behavioral support, or self-guided quit] in a 1:1:1 ratio. Due to the nature of the intervention, blinding was not done for the researcher or participant.

Arm 1: Participants in the NRT group were provided behavioral support and nicotine patches and/or either nicotine gum or lozenges based on their personal preference. The NRT quit plan was determined based on their Fagerstrom Test for Nicotine Dependence score modified for vaping [18,22].

Arm 2: Participants in the vape-taper group used their own ENDS and e-juice. They received both behavioral support from a pharmacist as well as a recommended nicotine vape-taper plan (Figure 1) based on their current e-juice nicotine concentration and vaping habits.

Arm 3: Participants in the self-guided group served as the control group and were asked to become vape-free and nicotine-free within 12 weeks. They did not receive behavioral support from the research team but were asked at each call and in-person appointment to discuss their quit attempt to help the researchers identify additional cessation methods for future studies.

Behavioral support provided to Arm 1 and Arm 2 was provided by the pharmacist after probing questions (Appendix A) to identify the participant’s motivating factors, challenges, and strategies for success. The pharmacist would provide some tips on strategies for success if the participant was unable to identify any. This time provided opportunity for open dialog between the participant and pharmacist.

Each participant received a $20 gift card at 4-week, 8-week, and 12-week appointments to cover study-related expenses such as time, travel, and phone usage. Participants in the self-guided group and vape-taper group were expected to purchase their own vaping supplies. Therefore, they received an additional $40 gift card at enrollment including at 4 weeks and 8 weeks to cover out-of-pocket expenses.

All participants had the same appointment and phone call schedule and were referred to the Michigan Tobacco Quitline [23].

Prospective data collection occurred over the six-month study period. Baseline data was collected from each participant at enrollment during the initial in-person appointment. Additional data was collected at phone calls at 3–7 days, 2 weeks, 6 weeks, 10 weeks, and 6 months. Participants were asked to attend in-person appointments at 4 weeks, 8 weeks, and 12 weeks to collect data, vitals, and receive NRT, if applicable. The Fagerstrom Test for Nicotine Dependence (FTND) was modified by the researchers (referred to as mFTND) to quantify nicotine addiction related to vaping habits [18,22]. The mFTND was used on all participants to track dependence over the study time period. All participants were asked a series of open-response questions based on standard prompts from the researchers related to withdrawal symptoms and other health effects related to the study. All participants were asked additional open-ended questions at the six-month call based on the standard prompts listed in Appendix A.

Participant demographics, tobacco use, and quit-method perceptions were summarized. Relative effectiveness of the three arms were measured by comparing the participant outcomes of successful vape and nicotine cessation at 12 weeks and 6 months. The percentage of participants who reported quitting at their 12-week and 6-month appointments was compared between groups using the chi-squared test. The Kruskal-Wallis test was used to compare mFTND between the three groups. Analysis of variance (ANOVA) was used to compare e-juice daily consumption and biometric measures (blood pressure, heart rate, and body weight) between groups. As a supplementary analysis, changes in the mFTND score and biometric measures from baseline to 12-week appointment were compared within each group using the Wilcoxon sign rank test and the paired t-test, respectively. Intent-to-treat analysis was completed on all study variables reported. When determining the status of quitting, any participant that was vaping or using nicotine products or were lost to follow-up was assumed to still be vaping. For other variables, missing data resulting from lost to follow-up or refusal to answer were excluded from the analysis of that variable. Test results with *p*-values less than 0.05 were considered significant. Open-response questions were recorded, coded, and then grouped into themes. Analysis of themes were conducted for every time the participant reported a symptom, barrier, benefit, or skill to overcome challenges. Symptoms were counted for all participants during each interaction. Participant-reported experiences related to symptoms were classified using a sentiment analysis to decipher positive and negative effects. Responses related to barrier, benefit, and skills were reported in results by the individual patient at the final appointment. One researcher did the initial coding that was then reviewed by another researcher. Disagreements were discussed with a third researcher to come to a conclusion.

The study was approved on 11 March 2019 by the Ferris State University Institutional Review Board under project identifier IRB-FY18-19-27.

## 3. Results

Twenty-nine individuals set up initial appointments, but initial data was only collected for 24 participants due to failure to attend the initial appointment (Appendix A). Of the 24 participants who were assigned to one of the study arms, eight participants were lost to follow-up after randomization. The NRT arm lost the highest percentage of participants compared to the vape taper arm and the self-taper arm, 42.9% vs. 25% and 33%, respectively. Twelve-week data was collected on 20 participants and six-month data was collected on 16 participants. The majority of our participants were male (71%) and white (79%). The average age of all study participants was 19.8 years ± 2.1. Mean age, sex, and vape history of each group are shown in Table 1. No statistical significance was seen in baseline demographics between groups. Participants in the NRT arm reported a lower concentration of nicotine e-juice and shorter duration of vaping. However, the NRT group reported more time spent vaping and consumed more milliliters of e-juice a day compared to the vape-taper and self-guided arms. These differences were not statistically significant (Table 1). The NRT arm had a lower average mFTND at baseline compared to the other arms (Table 1).

At baseline, the majority of our study participants had systolic blood pressures that were above the normal pressure of <120 mmHg (Table 1). The self-guided arm average systolic blood pressure decreased over 12 weeks, while the NRT and vape-taper groups had a modest increase in systolic blood pressure (Appendix A). At baseline, participants’ heart rates were within a normal range (60–100 bpm), and decreases were seen in most participants in the NRT and vape-taper groups (Appendix A). Increases in weight were seen in all arms, with the NRT arm having the largest average increase compared to the vape-taper and self-guided groups (Appendix A). No significant differences were found between groups for any vitals collected over the study period (Table 2).

Our primary endpoint was the number of participants to self-report being vape-free and nicotine-free at 12 weeks and 6 months. The self-guided arm had the highest percent of successful attempts with 77.8% of participants reporting as vape-free and nicotine-free at 12 weeks, but this number dropped to 44.4% at 6 months. The vape taper arm showed favorable results of 75% quit at 12 weeks and 6 months. The NRT arm had the lowest success rate with 42.9% at both time points (Table 2).

The mFTND score significantly decreased within all study arms from baseline to 12 weeks (Appendix A). The mFTND scores did not significantly differ between study arms at any data collection time point (Table 2). Mean e-juice daily consumption at 4 weeks and 8 weeks showed a significant difference between groups (Table 2). No participants used the Michigan Tobacco Quitline or any additional quit services during the study period.

Participants in all groups experienced positive and negative effects during the first 12 weeks while quitting. Appendix A categorizes the participant reported effects into positive or negative themes as a total count. The most reported positive effects over the entire study were improved concentration and focus, improved sleeping, more energy, and better breathing and exercise tolerance. One participant experienced significant skin improvement, specifically eczema clearing up with reduced ENDS usage. The most common reported negative effects from all treatment arms cumulatively were increased irritability or anger, increased appetite, difficulty sitting still or concentrating, and decreased sleep. Participants reported withdrawal symptoms to improve over time and had fewer cravings to vape or less satisfaction while vaping. The highest frequency of negative effects occurred in the first four weeks of the study and were seen most frequently in the self-guided arm (Appendix A).

NRT-specific adverse effects noted were itching and rash with the patch (*n* = 1) and worse heartburn with the gum (*n* = 1).

Participants in the behavioral support arms identified mindless use of ENDS as a significant barrier to quitting initially. Participants found it challenging to quantify ENDS use by amount (mL/day), time (h/day), and even triggers because it was such a subconscious activity. Participants noted that, after enrolling in the study, they were much more cognizant of when they were vaping, which allowed them to work on reducing their consumption.

At six months, 46% (*n* = 24) of participants reported the greatest challenge to quitting and remaining nicotine-free and vape-free was social pressure. Participants reported using self-control (29%, *n* = 24) and establishing new habits (21%, *n* = 24) as key techniques to overcome the pressures to vape. Of the 24 participants, 29% reported saving money and 20% reported feeling healthier as the greatest benefit to quitting.

## 4. Discussion

This study had a small sample size and was designed to be a pilot project to explore various methods of helping ENDS users to quit. This study was not powered to determine which method is superior or to provide an ENDS cessation protocol, but it shows that all methods are possibly effective options. The effects of different cessation approaches could be confounded by individual differences in pre-treatment nicotine habits, among other factors, such as differential dropouts. Differences in treatment methods could have caused implications to the study results. For example, the NRT arm received a smaller stipend since the researchers provided NRT and the other arms received a larger stipend to self-purchase ENDS supplies. The NRT arm also spent more time vaping and used more mL/day of e-juice at randomization compared to the other two arms. These measures could have been indicative of a stronger nicotine addiction, making it more difficult to quit. Additionally, the control arm was self-guided but were asked questions related to their quit attempt for the researchers to explore additional options for quit methods. The researchers knew that these questions could provide self-motivation but determined the possibility of additional quit strategies for future studies that outweighed those risks. Another limitation is self-reported quit success of ENDS and all tobacco, as the study did not incorporate biomarkers to confirm quit. This would be an important measure in future studies.

With smoking cessation research, it is common for quit-success rates to be low and usually <25% [24]. In this small sample size, quit rates at both 12 weeks and 6 months were seen at higher percentages of 42–77%. Many participants experienced minor slips or relapses during the 3-month observation period of the study, but this is very normal with smoking cessation and showed that the participants had the skills to get back to being quit. All arms had the same appointment schedule, similar data collection by the researchers, and were motivated to quit within two weeks of enrollment. This study is in alignment with Graham’s results supporting that the structure of a quit program is very important [16].

All study arms showed a decrease in mFTND scores, which is clinically significant to show that, as ENDS users consume lower concentrations of e-juice and vape less, they become less dependent on the habit of vaping and nicotine contained in the e-juice. The mFTND is not a validated tool and future research should establish its validity.

Participants of the study had more positive health benefits than anticipated. One benefit that is not in current literature, is skin improvement, specifically eczema clearing up for one participant. This may be an added benefit for users who are able to quit using ENDS. The majority of the other positive health benefits were improved sleeping, improved concentration, and breathing better. Positive benefits were seen most frequently in the vape-taper arm compared to the other two arms. Personal benefits that many of the users experienced included saving money, breaking dependence on nicotine, and pride in their accomplishment to quit. Negative health effects that were seen were expected issues with withdrawal symptoms and known adverse events of NRT. The self-guided arm experienced more negative effects compared to the arms with behavioral support from a pharmacist, indicating that support from a healthcare provider can make for a smoother quit attempt.

One of the barriers to quitting that participants identified through this study was the mindless use of ENDS, and participants stated that participating in a quit program allowed them to be more cognizant of when they were vaping in order to work on reducing their consumption. The NRT arm participants typically used less NRT than what was recommended by the pharmacist. This could have been a reason for the smaller percentage of participants in the NRT arm successful in quitting. The other large barriers with ENDS use are social factors and pressures. These are huge issues with young adults including our study population. Additional education is needed to change the perceptions and use of ENDS by taking away its “cool” persona [10].

This study population is not well representative of all ENDS users since it was conducted on a college campus and only students enrolled. The study also required participants to stop use of other tobacco products, but a 2018 publication from Hedman et al. reported 15% of ENDS users are dual users in Sweden [25]. Of the adult population, the age range with the highest percentage of ENDS users are 18–24-years-old, which does fall in line with our study population. The majority of the study population was white males, which is also supported as a group with the highest prevalence of e-cigarette use compared to females and other races [26,27,28].

While this study demonstrated that multiple methods may contribute to ENDS cessation success, and that pharmacist-led behavioral support is associated with ENDS cessation success, further research is warranted to better elucidate behavioral support strategies and pharmacotherapy that offer the greatest efficacy. Research should also focus on quantifying use and dependence in order to appropriately pair NRT. Some studies have been done to do this, but there are still many confounding factors with ENDS, such as nicotine concentration, lack of regulation on manufacturing and quality, quantifying use, device differences, and e-juice variances [29,30,31].

## 5. Conclusions

This study has described a successful and durable quit-program for some ENDS users. Participants who received behavioral support and a vape-taper from pharmacists were more likely to be vape-free and nicotine-free at six months. This study adds to the literature that frequent touchpoints and program structure are important pieces for helping motivated users to quit, and more robust assistance and behavioral support from pharmacists can provide a more positive quit attempt with fewer negative effects and a more lasting quit attempt.

## Figures and Tables

**Figure 1 pharmacy-09-00021-f001:**
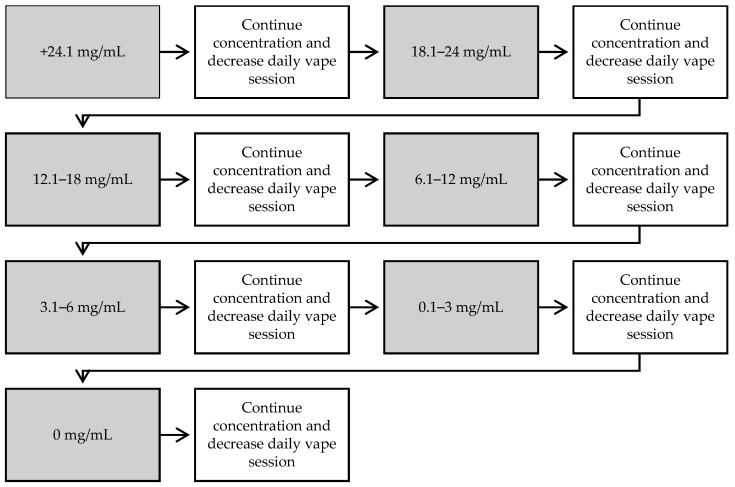
Taper schedule used for vape-taper arm. Vape taper aimed to decrease the amount of nicotine consumed by reducing concentration and frequency over time. Started at the participant’s current nicotine concentration of vape liquid. The first week the participant decreased vape exposure by one session per day or decreased the duration of sessions (~10–15% of time spent vaping), and, the second week, the participant decreased nicotine concentration of vape, as shown in the diagram above or about 20–25%, depending on available products. If unable to complete the step, then the same step was repeated until being successful before moving on. Steps were followed until the participant was vape-free and nicotine-free.

**Table 1 pharmacy-09-00021-t001:** Demographics and baseline data by the study arm.

Characteristic	Nicotine Replacement Therapy + Behavioral Support (*n* = 7)	Vape-Taper + Behavioral Support (*n* = 8)	Self-Guided (*n* = 9)
Mean Age (SD), y	22.6 (7.3)	20 (2.3)	19.4 (1.5)
Male, *n* (%)	5 (71.4)	7 (87.5)	5 (55.6)
Mean e-juice nicotine concentration (SD), mg/mL	40.14 (18.17)	43.13 (16.4)	49.11 (8.55)
Mean e-juice daily consumption (SD), mL/day	3.56 (6.33)	1.08 (0.5)	1.21 (0.91)
Mean time spent vaping (SD), h/day	2.64 (2.59)	0.95 (0.96)	1.3 (1.1)
Mean duration of vape history (SD), y	1.74 (1.31)	2.75 (1.28)	2.8 (2.33)
Mean past quit attempts (SD)	2.7 (1.1)	3.9 (3.5)	3 (1.3)
Mean mFTND score (SD)	4.57 (3.1)	5.38 (2.62)	6.11 (1.69)

mFTND = modified Fagerstrom Test for nicotine dependence.

**Table 2 pharmacy-09-00021-t002:** Selected results during the study period by each study arm.

Characteristic	Nicotine Replacement Therapy + Behavioral Support (*n* = 7)	Vape-Taper + Behavioral Support (*n* = 8)	Self-Guided (*n* = 9)	*p*-Value
Quit at 12 weeks ^1^, *n* (%)	3 (42.9)	6 (75)	7 (77.8)	0.280
Quit at 6 months ^1^, *n* (%)	3 (42.9)	6 (75)	4 (44.4)	0.350
Continuous quit at 6 months ^1,2^, *n* (%)	2 (28.6)	3 (37.5))	1 (11.1)	0.440
Mean mFTND score: 4 weeks (SD)	0.80 (1.79)	2.88 (1.86)	3.67 (2.78)	0.109
Mean mFTND score: 8 weeks (SD)	0.00 (0.00)	1.29 (1.60)	1.33 (1.41)	0.077
Mean mFTND score: 12 weeks (SD)	1.00 (1.41)	0.33 (0.82)	1.11 (2.26)	0.703
Mean mFTND score: 6 months (SD)	1.00 (2.00)	0.00 (0.00)	0.83 (1.60)	0.349
Mean e-juice daily consumption at 4 weeks (SD), mL/day	0.002 (0.004)	2.02 (3.27)	2.37 (2.50)	0.015
Mean e-juice daily consumption at 8 weeks (SD), mL/day	0.00 (0.00)	0.92 (1.40)	0.26 (0.59)	0.037
Mean e-juice daily consumption at 12 weeks (SD), mL/day	0.9 (1.75)	0.00 (0.00)	0.32 (0.71)	0.277
Mean e-juice daily consumption at 6 months (SD), mL/day	0.18 (0.35)	0.00 (0.00)	0.25 (0.52)	0.341
Mean Systolic blood pressure: 12 weeks (SD), mmHg	136.20 (13.04)	129.33 (13.16)	127.88 (6.94)	0.404
Mean heart rate: 12 weeks (SD), bpm	71.40 (17.60)	71.83 (12.83)	75.63 (19.62)	0.884
Mean Weight: 12 weeks (SD), lbs	205.80 (46.86)	185.00 (31.73)	171.88 (38.81)	0.338

^1^ Current use of ENDS or tobacco or lost-to-follow-up were analyzed as not quit. ^2^ Participants have not vaped or used any tobacco since the original quit date. mFTND = modified Fagerstrom test for nicotine dependence.

## Data Availability

The data presented in this study are available in Appendix A Statistics.

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
