# Peer review of "Pilot Study of Electronic Nicotine Delivery Systems (ENDS) Cessation Methods"

_pharmacy, 2021, doi:10.3390/pharmacy9010021_

Round 1

Reviewer 1 Report

Manuscript ID: pharmacy-1046863: Pilot Study of Electronic Nicotine Delivery Systems (ENDS) Cessation Methods; Sahr et al.  

This pilot study examined the differential efficacy of three different interventions for cessation of Electronic Nicotine Delivery Systems in adult users: i) nicotine replacement therapy (NRT) + behavioral support, ii) vape-taper + behavioral support and iii) self-guided. The authors report finding that at 12 weeks 42.9%  of those in i), 75% of those in ii), and 77.8% in iii) self-reported being vape- and nicotine-free. At 6 months, 42.9% of those in i), 75% in ii) and 44.4% in iii) self-reported being vape- and nicotine-free, with those who received behavioral support and a vape-taper plan were most likely to be vape- and nicotine-free at 6 months, and participants identified self-control and establishing new habits as important strategies stop vaping.

This study represents an important effort to better address treatment strategies for individuals trying to stop vaping. I have some questions/comments that might help to clarify the manuscript and its conclusions:

1. Were users of other forms of nicotine containing products excluded from the study, e.g., those using traditional combustible cigarettes? What was the history of tobacco use and prior treatments in participants and how is this controlled for in the study/analyses?

2.  Analyses did not suggest that the groups significantly differed from each other in terms of self-reported vaping cessation or change in e-juice consumption at 12-weeks or 6-months, while from the results listed in supplemental table 2 all participants appeared to experience a significant reduction in nicotine dependence at 12-weeks. Can the authors please clarify this throughout the manuscript including in the abstract?

3.  Given the effect size derived from this study, what sample size would be needed for the authors to detect a significant difference in vaping cessation across the treatment groups?

4.  Supplemental table 3 is confusing, do the numbers in the table represent the frequency (%) of individuals that endorsed any of the symptoms in that category during that timeframe and the counts are in the footnote?

5.  Could the authors describe in more detail how the open-ended questions/responses were asked, coded and analyzed? Many of these results seem to be introduced for the first time in the discussion (e.g., some barriers to quitting).

6.  Likewise, other things reported in the abstract and results are not included in the discussion, can the authors please be more consistent with this?

7.  Why are the authors focusing on 12-weeks findings at times and then at other times the 6-month findings? Can they please report the number of participants in each group at 6-months in Supplemental Figure 1? These numbers at 6-months need to be included in main manuscript as well.

8.  Since the authors do focus on the 6-month results, how did the individuals that dropped out by 6-months differ from sample retained?

9.  Can the authors clarify that they are running intent-to treat analyses based on N=24? However, the open-ended questions at 6-months can only focus on those who completed (N=?)?

10. Did the authors include any biomarkers of nicotine abstinence, e.g., saliva or blood nicotine levels?

Author Response

Reviewer 1: This pilot study examined the differential efficacy of three different interventions for cessation of Electronic Nicotine Delivery Systems in adult users: i) nicotine replacement therapy (NRT) + behavioral support, ii) vape-taper + behavioral support and iii) self-guided. The authors report finding that at 12 weeks 42.9%  of those in i), 75% of those in ii), and 77.8% in iii) self-reported being vape- and nicotine-free. At 6 months, 42.9% of those in i), 75% in ii) and 44.4% in iii) self-reported being vape- and nicotine-free, with those who received behavioral support and a vape-taper plan were most likely to be vape- and nicotine-free at 6 months, and participants identified self-control and establishing new habits as important strategies stop vaping.

This study represents an important effort to better address treatment strategies for individuals trying to stop vaping.

Author: Thank you for your comments and thoughtful review to improve our paper.

 I have some questions/comments that might help to clarify the manuscript and its conclusions:

  1. Were users of other forms of nicotine containing products excluded from the study, e.g., those using traditional combustible cigarettes? What was the history of tobacco use and prior treatments in participants and how is this controlled for in the study/analyses?

Participants were asked to solely vape during study. Added to Lines 63-64

Additional history of tobacco use beyond what was reported and prior treatments were not collected

Participants were randomly assigned to each arm to control for differences of participants and no statistical differences were found between groups as presented in Table 1

  1. Analyses did not suggest that the groups significantly differed from each other in terms of self-reported vaping cessation or change in e-juice consumption at 12-weeks or 6-months, while from the results listed in supplemental table 2 all participants appeared to experience a significant reduction in nicotine dependence at 12-weeks. Can the authors please clarify this throughout the manuscript including in the abstract?

Lines 108-114: Added details to methods for primary and supplemental analyses.

Lines 178-183 added to clarify the reviewer’s question. Table 2 p-value is comparing all three groups for specific measures listed in the first column. No statistical differences were found BETWEEN groups for any variable.

Supplemental Table 2 are analyses WITHIN a group comparing baseline to 12-week. Supplement table 2 shows the significant difference in mFTND from baseline to 12 weeks suggesting a decrease in nicotine dependence. This correlates with data in table 2 showing a decrease in e juice consumption over the study period. It also correlates with level of quit success within the Taper arm having the highest success and lowest e-juice consumption and mFTND score.

  1. Given the effect size derived from this study, what sample size would be needed for the authors to detect a significant difference in vaping cessation across the treatment groups?

The cessation success rate between compared groups (our primary end point) were not significantly different (p<0.05), and for this reason, the authors are hesitant to quantify the effect size. While it is sometimes argued that p-values and effect sizes are separate concepts, we believe it should be considered together in our case. Based on the result that our p-values are substantially large and that it is quite uncertain one method is superior to other methods, we would like to take extra caution and avoid suggesting any magnitude of “effect” in our paper. That being said, if any reader wishes to compute the effect size, we report the actual number of participants who succeeded cessation. Since it is a binary outcome, the computation of mean and standard deviation to derive effect size would be possible.

In the future investigations, the determination of minimum sample size should take into account an appropriate level of significance and power, in addition to the allocation ratio between treatment and control groups.

  1. Supplemental table 3 is confusing, do the numbers in the table represent the frequency (%) of individuals that endorsed any of the symptoms in that category during that timeframe and the counts are in the footnote?

Supplemental Table 3 are number of times participants reported each effect over representative time period in each study arm. Title was revised to “Counts” instead of frequencies to more accurately reflect what the numbers stand for, and in the footnote are total counts for each symptom across all groups. Also, added footnote information to Supplemental Table 3 to clarify the time points for when data was collected and number of times it was collected.

Lines 119-124-added coding information to methods to help explain

Lines 186-192- updated to clarify open-response information shown in Supplemental Table 3

  1. Could the authors describe in more detail how the open-ended questions/responses were asked, coded and analyzed? Many of these results seem to be introduced for the first time in the discussion (e.g., some barriers to quitting).

Lines 100-102: Open-ended questions were asked based on question prompts below:

  1. What changes in your mood have you experienced since quitting vaping?
  2. What changes have you experienced in your appetite or sleeping?
  3. What changes have you experienced in your ability to concentrate or sit still?
  4. What other changes in your health have you noticed since quitting vaping/ changing vaping habits?
  5. Have you experienced dizziness, headache, or a skin rash where your NRT patch is applied?
  6. What other symptoms have you experienced since quitting vaping?

If yes to any questions above- Follow up when it started/stopped.

The authors did not feel these questions were necessary to the manuscript but can be incorporated if the reviewers/editor feel it would add value to the manuscript.

Lines 82-87: Discussed how the behavioral support open-ended questions and responses were administered.

Lines 102-104 and S1 Table footnotes: Open-response questions aimed at behavioral support are provided in the Supplemental Table 1 and were asked to participants in both behavioral support arms for all questions. 6 month questions were asked to all participants.

Lines 119-124: Additional details were added to discuss coding and analysis.

Lines 199-203: discussed open-ended questions’ results related to barriers and benefits

  1. Likewise, other things reported in the abstract and results are not included in the discussion, can the authors please be more consistent with this?

Thank you for bringing this to our attention as this was not our intention. The authors have reviewed these sections to make sure there is consistency.

Lines 208-220 were added to the discussion to highlight additional limitations from study methods and population

Lines 229-231: added to discuss mFTND score decreases within each arm

  1. Why are the authors focusing on 12-weeks findings at times and then at other times the 6-month findings? Can they please report the number of participants in each group at 6-months in Supplemental Figure 1? These numbers at 6-months need to be included in main manuscript as well.

The authors have updated the Supplemental Figure 1 to reflect the 6 month data as well.  Lines 131-136 have been updated to clarify number of participants at selected time-points. A larger focus has been put on 12 week information since this was the active study period with frequent touchpoints, whereas 6 month information was an observation period to collect additional data.

  1. Since the authors do focus on the 6-month results, how did the individuals that dropped out by 6-months differ from sample retained?

The authors did not further investigate the participants who were lost to follow-up since the lost to follow up was stratified across all groups. To side on the error of caution, all participants lost to follow-up were considered to still be vaping even though individuals may have dropped out due to successfully quitting and not see the need to continue appointments or calls so our success rate is probably lower than reality.

  1. Can the authors clarify that they are running intent-to treat analyses based on N=24? However, the open-ended questions at 6-months can only focus on those who completed (N=?)?

Lines 114-115 were added to clarify that Intent-to-treat analysis was completed on all participants who were randomized (n=24)

Lines 119-124 were updated. Open-ended responses were reported by patients at each appointment. Symptoms were reported as number of times participants reported them at unique encounters, but the same participant could have given the same symptom 5 times at different encounters and it was reported as 5. For benefits, challenges, and skills that were reported as percentages, these were unique patients at the final appointment. Percentages were based on n=24 and Lines 199-203 have been updated to reflect intent-to-treat n=24. Originally, these percentages were based on n=16 for those who completed the 6 month call, but the authors feel it is better to under-report these numbers than over-report.

  1. Did the authors include any biomarkers of nicotine abstinence, e.g., saliva or blood nicotine levels?

No, since this was a pilot study the authors did not feel that biomarkers were necessary at this study level but would be important to include in larger studies intended to impact clinical guidelines. The participants were not incentivized by quit status so there was no reason for participants to feel pressured to lie about quit status. This has been added as a limitation to the discussion.

Reviewer 2 Report

The manuscript by Sahr et al reported results from a pilot study which compared three cessation approaches among ENDS users who wanted to quit. The results provide support for continued large-scale research on the promising cessation methods. While overall the study methods and findings were well presented, I suggest that the authors carefully address some remaining issues so that it can be considered for publication.

Although the authors briefly acknowledged the study was under powered to yield meaningful comparison of the efficacy of the three approaches, a fuller discussion of the limitations with this design feature is warranted given the risk for the results and implications to be misinterpreted by some readers. For example, relative to the other two groups, the NRT group at pre-cessation baseline tended to be vaping more intensively despite using of low-nicotine e-liquids and it is unclear whether and how the pre-cessation individual differences affected cessation outcomes. With a large sample size, such pre-existing differences are likely to be evened out by randomization. In a small sample size study like this one, however, the effects of different cessation approached could be confounded by individual differences in pre-treatment among other factors such as differential drop-outs.

The intervention components and how they may account for the effectiveness for aiding cessation seem to have some nuances worthy of discussion to help readers understand the implications of the results. For example, in the self-guided group, a component was asking the participants to discuss participants’ quit attempt to help the researchers identify additional cessation methods for future studies. The unique process could have served a motivator for the participants to be more committed to cessation through self-validation or positive impression management mechanisms. Another example is that the NRT group did not received gift cards like the other two groups did. The difference in reinforcement could also have contributed to treatment outcome differences. These finer points should be discussed as they are relevant to mechanisms of action for each of the treatments, besides the intended primary treatment elements. In general, the paper can be significantly improved by better discussion of implications of the results as to the rationales or mechanisms of action of the cessation interventions.

Vaping is often related to cigarette smoking (e.g., dual users or vaping users being ex-smokers). Participants’ smoking behavior and smoking history prior to vaping cessation were not reported, but these could be important factors to influence vaping cessation outcome. The authors should address this either by looking at the smoking data if available or discuss the limitations of not considering impact of smoking in this study.

The primary endpoints are self-reported vaping and nicotine use status. The weaknesses of self-reported outcome, relative to bio-chemically verified abstinence from the use of nicotine products (e.g., cotinine or CO levels) should be discussed.

As negative effects of vaping cessation are likely to be interesting to readers, they should be, at least briefly, reported in Results, rather than only being referred to Supplemental Materials.

Some minor issues should also be addressed.

Line 11: The 2nd “currently” in the sentence should be removed.

The future tense is used in the caption of Figure 1. Instead, the past sense is more appropriate.

Line 189: “data collection from the researchers” should be replaced with “data collection by the researchers”?

Line 198: “dependence to nicotine” should be “dependence on nicotine”?

Line 212: “does not correlate to” should be replaced with “is not well representative of”?

Line 225: The first sentence in the conclusion may be rewritten to reflect that not all users were successful in cessation.

Author Response

Reviewer #2: Comments and Suggestions for Authors

The manuscript by Sahr et al reported results from a pilot study which compared three cessation approaches among ENDS users who wanted to quit. The results provide support for continued large-scale research on the promising cessation methods. While overall the study methods and findings were well presented, I suggest that the authors carefully address some remaining issues so that it can be considered for publication.

Author: Thank you for your review and helpful comments!

Although the authors briefly acknowledged the study was under powered to yield meaningful comparison of the efficacy of the three approaches, a fuller discussion of the limitations with this design feature is warranted given the risk for the results and implications to be misinterpreted by some readers. For example, relative to the other two groups, the NRT group at pre-cessation baseline tended to be vaping more intensively despite using of low-nicotine e-liquids and it is unclear whether and how the pre-cessation individual differences affected cessation outcomes. With a large sample size, such pre-existing differences are likely to be evened out by randomization. In a small sample size study like this one, however, the effects of different cessation approached could be confounded by individual differences in pre-treatment among other factors such as differential drop-outs.

Added to Discussion Lines 208-220 on limitations: The effects of different cessation approaches could be confounded by individual differences in pre-treatment among other factors such as differential dropouts.

The intervention components and how they may account for the effectiveness for aiding cessation seem to have some nuances worthy of discussion to help readers understand the implications of the results. For example, in the self-guided group, a component was asking the participants to discuss participants’ quit attempt to help the researchers identify additional cessation methods for future studies. The unique process could have served a motivator for the participants to be more committed to cessation through self-validation or positive impression management mechanisms. Another example is that the NRT group did not received gift cards like the other two groups did. The difference in reinforcement could also have contributed to treatment outcome differences. These finer points should be discussed as they are relevant to mechanisms of action for each of the treatments, besides the intended primary treatment elements. In general, the paper can be significantly improved by better discussion of implications of the results as to the rationales or mechanisms of action of the cessation interventions.

Lines 208-220: Added these points to the discussion as they are pertinent to readers and important for clinicians to weigh when considering treatment.

Vaping is often related to cigarette smoking (e.g., dual users or vaping users being ex-smokers). Participants’ smoking behavior and smoking history prior to vaping cessation were not reported, but these could be important factors to influence vaping cessation outcome. The authors should address this either by looking at the smoking data if available or discuss the limitations of not considering impact of smoking in this study.

 Researchers did not collect a detailed history on past smoking or tobacco history and focused more on past attempts as presented in table 1. The participants were asked about concurrent tobacco use for purposes of asking patients to only use e-cigs or be excluded from the study. Lines 30-31 & 255-256 were added with the most recent U.S. data for 18-24 year old persons with dual use since the researchers were unable to find data of cigarette history for current e-cig users.

The primary endpoints are self-reported vaping and nicotine use status. The weaknesses of self-reported outcome, relative to bio-chemically verified abstinence from the use of nicotine products (e.g., cotinine or CO levels) should be discussed.

 Added to lines 219-220 for limitations of this study

As negative effects of vaping cessation are likely to be interesting to readers, they should be, at least briefly, reported in Results, rather than only being referred to Supplemental Materials.

 Lines 190-192 now include the most common negative effects seen

Some minor issues should also be addressed.

Line 11: The 2nd “currently” in the sentence should be removed.

Removed from abstract line 11

The future tense is used in the caption of Figure 1. Instead, the past sense is more appropriate.

Caption for Figure 1 (lines 147-153) were updated to past tense

Line 189: “data collection from the researchers” should be replaced with “data collection by the researchers”?

updated as suggested

Line 198: “dependence to nicotine” should be “dependence on nicotine”?

updated as suggested

Line 212: “does not correlate to” should be replaced with “is not well representative of”?

updated as suggested

Line 225: The first sentence in the conclusion may be rewritten to reflect that not all users were successful in cessation.

Line 269 was slightly modified to more accurately portray the study results

Round 2

Reviewer 1 Report

The authors responded well to reviewer comments.